# *SPON1* Can Reduce Amyloid Beta and Reverse Cognitive Impairment and Memory Dysfunction in Alzheimer’s Disease Mouse Model

**DOI:** 10.3390/cells9051275

**Published:** 2020-05-21

**Authors:** Soo Yong Park, Joo Yeong Kang, Taehee Lee, Donggyu Nam, Chang-Jin Jeon, Jeong Beom Kim

**Affiliations:** 1Stem Cell Research Center, School of Life Sciences, Ulsan National Institute of Science and Technology (UNIST), Ulsan 44919, Korea; shorypark@unist.ac.kr (S.Y.P.); kangj0594@unist.ac.kr (J.Y.K.); thlee525@hotmail.com (T.L.); namdongg@unist.ac.kr (D.N.); 2Neuroscience Laboratory, Department of Biology, College of Natural Sciences, Kyungpook National University, Daegu 41566, Korea; cjjeon@knu.ac.kr

**Keywords:** Alzheimer’s disease, stem cell-based gene therapy, gene therapy, *SPON1*, amyloid beta, HEK 293T cells, beta-secretase, induced neural stem cells

## Abstract

Alzheimer’s disease (AD) is a complex, age-related neurodegenerative disease that is the most common form of dementia. However, the cure for AD has not yet been founded. The accumulation of amyloid beta (Aβ) is considered to be a hallmark of AD. Beta-site amyloid precursor protein cleaving enzyme 1 (BACE1), also known as beta secretase is the initiating enzyme in the amyloidogenic pathway. Blocking BACE1 could reduce the amount of Aβ, but this would also prohibit the other functions of BACE1 in brain physiological activity. SPONDIN1 (SPON1) is known to bind to the BACE1 binding site of the amyloid precursor protein (APP) and blocks the initiating amyloidogenesis. Here, we show the effect of *SPON1* in Aβ reduction in vitro in neural cells and in an in vivo AD mouse model. We engineered mouse induced neural stem cells (iNSCs) to express *Spon1*. iNSCs harboring mouse *Spon1* secreted SPON1 protein and reduced the quantity of Aβ when co-cultured with Aβ-secreting Neuro 2a cells. The human *SPON1* gene itself also reduced Aβ in HEK 293T cells expressing the human *APP* transgene with AD-linked mutations through lentiviral-mediated delivery. We also demonstrated that injecting *SPON1* reduced the amount of Aβ and ameliorated cognitive dysfunction and memory impairment in 5xFAD mice expressing human *APP* and *PSEN1* transgenes with five AD-linked mutations.

## 1. Introduction

Alzheimer’s disease (AD) is the most prevalent age-related neurodegenerative disease [1]. There have been many attempts to cure AD [2] but, definitive treatment has yet to be developed [3]. Amyloid beta (Aβ) and hyperphosphorylated tau are major pathological hallmarks of Alzheimer disease. Several studies showed that inflammatory reactions or neuronal cell death in the AD patient’s brain occur due to high Aβ accumulation as well as tau pathology [4,5,6,7]. Therefore, studies have focused on reducing the amount of Aβ in the brain to cure AD [8].

One possible therapeutic approach is to target the enzymes that process amyloidogenesis [9]. Aβ is derived from the amyloid precursor protein (APP) through sequential cleavages by beta-site amyloid precursor protein cleaving enzyme 1 (BACE1) and γ-secretase [10,11]. Reduction of Aβ through inhibition of BACE1 has been studied as a way to find a treatment for AD [12,13,14]. However, BACE1 not only initiates amyloidogenesis, but also conducts other functions in brain physiological activity, such as processing sodium current, synaptic transmission, and myelination [15,16,17]. Thus, the inhibition of BACE1 could cause dysfunctions in the brain physiological activities [18,19,20].

To overcome this limitation, SPONDIN1 (SPON1), also known as F-Spondin has been studied as a novel therapeutic approach. SPON1 is an extracellular matrix protein, consisting of three domains: spondin, reelin, and thrombospondin, and plays an essential role in neuronal survival and the migration of commissural axons [21,22]. SPON1 binds to the BACE1 cleavage site of APP and prohibits APP from being cleaved and released as the Aβ form [23]. SPON1 also interacts with the apolipoprotein E receptor ApoEr2 and clusters with APP, resulting in decreased production of Aβ [24]. Even though SPON1 has the potential to inhibit the amyloidogenic pathway, it has not been widely studied for AD therapy [25].

To confirm the efficacy of AD therapy, testing on neural cells isolated from patients may show the efficacy more accurately than testing on neural cell lines [26,27]. However, neural cells are difficult in isolation from the brain and in large-scale expansion for drug testing [28]. Recently, a direct lineage conversion was established to generate neural stem cells which proliferate well from fibroblasts by ectopic expression of transcription factors [29,30].

In this study, we generated mouse induced neural stem cells by the ectopic expression of *Octamer-binding transcription factor 4* (*Oct4*) and then genetically engineered them to express mouse *Spon1* (iNSC-Spon1). When iNSC-Spon1 was co-cultured with mouse neuroblastoma cells, Neuro 2a cells (N2a) which secrete Aβ, the amount of secreted Aβ was reduced. Next, we administered lentiviral vector encoding human *SPON1* into human HEK-293T, which were engineered to express human *APP* with AD-linked mutations, and confirmed that human *SPON1* reduced the secretion of Aβ. To evaluate the effect of human *SPON1* gene in mice expressing human *APP* and *PSEN1* transgenes with five AD-linked mutations (5xFAD), lentiviral-human *SPON1* was injected into the hippocampus and entorhinal cortex (EC) in the early stage of amyloidogenesis. The amount of Aβ was reduced, and memory and cognition of 5xFAD mice were improved after human *SPON1* injection. Therefore, we showed that *SPON1* may be used as a treatment for AD.

## 2. Materials and Methods

### 2.1. Cell Culture

Mouse primary dermal fibroblasts were isolated from the back skin of female CF-1 mice (6-week-old, 16–18 g, ORIENT, Seongnam, Gyeonggi, South Korea), as described previously [31]. The mice were anesthetized with 1–2% isoflurane. Dorsal mouse skin (60 × 100 mm) was harvested using scissors and minced using razor blades. After incubation with collagenase IV (1 mg/mL) at 37 °C for 1 h, the cell suspension was passed through a 70-μm filter. Primary skin fibroblasts were cultured in DMEM medium (Gibco, Grand Island, NY, USA) supplemented with 10% fetal bovine serum (FBS) (Gibco, Grand Island, NY, USA) and 1% penicillin/streptomycin (Gibco, Grand Island, NY, USA).

iNSCs were derived from mouse skin fibroblasts by transduction of retroviral vectors encoding mouse *Oct4* as previously described [30]. Skin fibroblasts (3 × 10^4^ per well) were seeded on a 6-well plate and incubated with retrovirus containing mouse *Oct4* supernatants. Three days after infection, the medium was replaced with NSC medium, comprising DMEM/F12 medium (Gibco, Grand Island, NY, USA) supplemented with N2 (Gibco, Grand Island, NY, USA), 2 mM L-Glutamine (Gibco, Grand Island, NY, USA), 1% penicillin/streptomycin and 10 ng/mL each of FGF-2 (Peprotech) and EGF (Peprotech). The medium was replaced every 2–3 days.

N2a cells were gifted by Prof. Jeon, Chang-Jin (Kyungpook National University, Daegu, South Korea) and cultured in DMEM medium (Gibco, Grand Island, NY, USA) supplemented with 10% FBS (Gibco, Grand Island, NY, USA) and 1% penicillin/streptomycin (Gibco, Grand Island, NY, USA). HEK 293T cells were purchased from American Type Culture Collection (ATCC, Manassas, Virginia, USA) and cultured in DMEM medium (Gibco, Grand Island, NY, USA) supplemented with 10% FBS (Gibco, Grand Island, NY, USA) and 1% penicillin/streptomycin (Gibco, Grand Island, NY, USA). All cells were maintained in a humidified atmosphere containing 5% CO_2_ at 37 °C.

### 2.2. Vector Construction

To construct the lentiviral vector encoding *Spon1*, mouse *Spon1* cDNA sequence encoding amino acid 1–807 for full length *Spon1* was amplified by PCR. Human *SPON1* cDNA sequence encoding amino acid 1–807 for full length *SPON1* was amplified by PCR. These mouse *Spon1* and human *SPON1* PCR products were ligated to a pLVX lentiviral vector encoding the hygromycin resistance gene (pLVX-Spon1 and pLVX-SPON1 respectively).

To construct the lentiviral vector encoding human *APP* with AD-linked mutations, the coding sequence of the human *APP* transgene with Swedish (K670N/M671L), Florida (I716V), and London (V717I) mutations was amplified from 5xFAD tail gDNA by PCR. The PCR product was ligated to pLVX lentiviral vector encoding the puromycin resistance gene (pLVX-APP).

All viral vectors were packaged in HEK 293T cells using x-tremeGENE9 (Roche, Mannheim, Baden-Württemberg, Germany) according to the manufacturer’s protocols.

### 2.3. Therapeutic Engineering

iNSCs (5 × 10^4^ cells/well on a 12-well plate) were plated. The next day, the cells were treated with 4 μg/mL protamine sulfate and then incubated with pLVX-Spon1 for 24 h. The infected cells were selected by 200 μg/mL hygromycin treatment for 2 weeks (iNSC-Spon1).

HEK 293T cells (5 × 10^4^ cells/well on a 12-well plate) were plated. The next day, the cells were treated with 4 μg/mL protamine sulfate and then infected with pLVX-APP for 24 h. The infected cells were selected by 1 μg/mL puromycin treatment for 2 weeks (293T-APP).

The integration of the transgene was confirmed by PCR-based genotyping. The cells (1 × 10^6^ cells) were lysed with 75 μL 50% NaOH at 98 °C for 1 h. and neutralized with 75 μL 40mM Tris-HCl. PCR was performed with 2 uL cell lysate, Taq DNA polymerase mixture (Invitrogen, Carlsbad, CA, USA) and primers. The primer sequences are listed in Appendix A. PCR was performed in the following steps; 94 °C 180 s (pre-denaturation), 94 °C 45 s (denaturation), 57 °C 30 s (annealing), 72 °C 90 s (elongation), 72 °C 600 s (final elongation). The total cycle number from denaturation to elongation was 35.

### 2.4. Conventional and Quantitative Real-Time PCR

RNA was extracted by an RNA mini kit (Qiagen, Hilden, North Rhine-Westphalia, Germany) through the manufacturer’s protocol. RNA was converted into cDNA by MMuLV-reverse transcriptase (NEB, Ipswich, MA, USA), and 1 μL cDNA was added with SYBR green (Bio-Rad, Hercules, CA, USA) for qRT-PCR. cDNA was added with Taq DNA polymerase for conventional PCR. The primer sequences are listed in Appendix A.

### 2.5. Immunocytochemistry

iNSC-Spon1 and 293T-APP-SPON1 cells were fixed by 4% paraformaldehyde for 15 min, and then 1% Triton-X was treated for permeabilization. Next, cells were incubated with 4% FBS for 30 min. Primary antibodies that were diluted by 4% FBS were added for 1 h at room temperature (RT). After washing, secondary antibodies (1:1000) diluted in PBS were added for 1 h at RT. Secondary antibodies included Alexa488 anti-goat antibody (ab150129, abcam, Cambridge, UK), which was used for V5 and Sox2 staining, and Alexa555 anti-mouse antibody (ab150118, abcam, Cambridge, UK), which was used for Nestin staining. Cells were counterstained by Hoechst 33,342 (Invitrogen, Carlsbad, CA, USA). The cells were imaged using a Leica fluorescence microscopy (DMI 3000 B, Leica, Wetzlar, Hesse, Germany). The primary antibodies are listed in Appendix A.

### 2.6. ELISA

To quantify the secretion of mouse SPON1, 2 × 10^5^ cells of iNSCs or iNSC-Spon1 were seeded in one well of 6well plate and the media was harvested after 48 h. Mouse SPON1 ELISA (Cusabio, Wuhan, Hubei, China) was performed according to the manufacturer’s protocol.

To verify Aβ reduction by iNSC-Spon1 through a co-culture system, 2 × 10^5^ cells of iNSCs or iNSC-Spon1 and 2 × 10^5^ cells of N2a were seeded on the same well of a 6-well plate for 48 h. The media was harvested and Aβ40 and Aβ42 ELISA assays (Invitrogen, Carlsbad, CA, USA) were performed according to the manufacturer’s protocol.

To verify Aβ reduction by human *SPON1*, 2 × 10^5^ cells of 293T and 293T-SPON1 were seeded in one well of a 6-well plate and the media were harvested after 24 h. Aβ40 and Aβ42 ELISA (Wako, Chuo-Ku, Osaka, Japan) were performed according to the manufacturer’s protocol.

All ELISA assays were performed in biological triplicate.

### 2.7. In Vivo Experiments

All animal studies were approved by Ulsan National Institute of Science Institutional Animal Care and Use Committee (UNISTIACUC-18-04) and carried out in accordance with the Ulsan National Institute of Science and Technology animal guideline. Animals were housed in an animal care cage (4–5 per cage) under standard environmental conditions (temperature 20 ± 2 °C; humidity 40 ± 5%; 12 h light/dark cycle) and were provided with normal pellet food and water ad labium. 5xFAD mice were purchased from The Jackson Laboratory (Bar Harbor, ME, USA) and were maintained on a hybrid C57BL6/SJL background. All surgeries were carried out with sterilized surgical instruments in the clean animal surgery room of the UNIST in vivo research center (Ulsan, South Korea) during the daytime. Human *SPON1* virus or mKatet2 virus was generated as previously described [32]. The pLVX-SPON1 or pLVX-mKate2 vectors were transfected into HEK cells (5 × 10^6^ cells per plate of a 100-mm plate/total 12 plates). Two days after transfection, the medium was harvested and spun down twice at 80,000× *g* for 1.5 h. Virus pellets were resuspended by 130 μL PBS. For surgery, 5xFAD female mice (three months old, 19–22 g, *n* = 14) were fixed in a stereotaxic frame (Stoelting, Wood Dale, IL, USA) under anesthesia with 1–2% isoflurane. The scalp of the mouse was excised and four holes were drilled in the skull according to coordination (hippocampus: AP: −2.02, ML: ±1.8; entorhinal cortex AP: −4.7, ML: ±2.9). Human *SPON1* virus or mKatet2 virus was stereotactically injected into the hippocampus (AP: −2.02, ML: ±1.8, DV: −1.9) [33] and entorhinal cortex (AP: −4.7, ML: ±2.9, DV: −3.1) [34] at a rate of 0.5 μL/min using a Hamilton microsyringe and motorized stereotaxic injector (Stoelting, Wood Dale, IL, USA). Non-Tg wild-type littermates received a sham surgery as control.

### 2.8. Morris Water Maze Assay

The Morris water maze (MWM) assay was performed to examine hippocampal-dependent learning and memory following the standard protocol 6 months after injection [35,36]. We used a water bath (diameter of 95 cm). The water temperature was 26 ± 1 °C. White paint was released in the water to make the platform (target) invisible to the mice. We attached four different cues on the interior of the pool at the location of North, South, East, and West above the water surface. The training for the hidden platform of the MWM assay consisted of four trials each day for 5 days. The mouse start position is listed in Appendix A. The location of the platform was fixed in the South West quadrant. For the 5 days of training, the mice were trained to find a submerged platform. The training time for 1 trial was 1 min, and the interval between trials was 15 min. If the mouse found the platform before this time, the time was recorded. The probe trial was performed without the platform for 1 min 24 h after the last day of the training. During the trials, the mouse’s movement was tracked and analyzed by SMART software (Version 3.0, Panlab Harvard Apparatus, Barcelona, Spain).

### 2.9. Western Blot

Mice were sacrificed by intraperitoneal injection of avertin (300 μL per 10 g of mouse weight) generated by mixing 2,2,2-tribromoethanol:Tert-amyl alcohol (1:1 ratio) and perfused with PBS after MWM assay. The brain was extracted and divided into two halves: the right and left cerebral hemispheres. The right hemisphere was used for western blot. The hippocampus from the right hemisphere was mechanically dissected. Tissues were homogenized as previously described [37]. The hippocampus was weighted, and 5 times the volume of tissue homogenization buffer (tissue weigh: buffer volume = 1:5) consisting of 2 mM Tris (Bio-Rad, Hercules, CA), 250 mM sucrose (Sigma-aldrich, St. Louis, MO, USA), and 0.5 mM EDTA (Sigma-aldrich, St. Louis, MO, USA), 0.5 mM EGTA (Sigma-aldrich, St. Louis, MO, USA) with a protease inhibitor cocktail (1:100, Sigma-aldrich, St. Louis, MO, USA) was added into tissues and mechanically homogenized by TissueLyser LT (Qiagen, Hilden, North Rhine-Westphalia, Germany). The same volume of 0.4% diethylamine (Sigma-aldrich, St. Louis, MO, USA) was added into the homogenate and it was sonicated it for 1 min. The homogenates were spun down at 100,000× *g* for 1 h at 4 °C. The supernatant was harvested and quantified by BCA protein assay. The supernatant (20 μg) was loaded onto an 8–15% gradient polyacrylamide gel and transferred onto an immune-blot PVDF membrane. The membrane was boiled in PBS for 5 min and incubated in a blocking solution (6% skim milk, Bio-Rad, Hercules, CA, USA) for 1 h at 4 °C. The membrane was incubated with the primary antibodies diluted in the blocking solution for 16 h at 4 °C. After washing, the membrane was incubated with HRP-conjugated secondary antibodies diluted in the blocking solution. Finally, all protein bands were visualized using chemiluminescence substrates. The antibodies were listed in Appendix A. The images were analyzed by ImageJ software. We selected the rectangle tool in ImageJ and drew a frame around the band as a region of interest (ROI). The density of every band was measured in the same sized frame. We calculated the ratio of the density of the Aβ oligomer and HSC-70 ROI.

### 2.10. Immunohistochemistry

The left hemisphere of the mice sacrificed after the behavior test was used for immunohistochemistry. The left hemisphere was fixed with 4% paraformaldehyde for 24 h at 4 °C, incubated with 30% sucrose for 48 h 4 °C and then frozen at −80 °C after embedding in optimal cutting temperature (OCT) compound (Sigma-aldrich, St. Louis, MO, USA) overnight. The specimen was cryosectioned at a thickness of 30 μm using Cryostat (CM1950, Leica, Buffalo Grove, IL, USA). The sections were mounted on a saline-coated slide glass. For antigen retrieval, the slide was treated with 0.1 M sodium citrate with 0.05% Tween-20 and boiled at 120 °C for 10 s and at 90 °C for 10 min. The slide was cooled at RT for 30min. Then, the slide was permeabilized with 1% triton-X (Sigma-aldrich, St. Louis, MO, USA) for 10 min and then blocked with 5% FBS for 30min. Primary antibodies were diluted in 5% FBS and treated in the sections overnight at 4 °C. After washing, secondary antibodies diluted in PBS (1:1000) were treated for 1 h at RT. Secondary antibodies included Alexa488 anti-goat antibody (ab150129, abcam, Cambridge, England, UK), which was used for V5 staining, and Alexa488 anti-mouse antibody (ab150118, abcam, Cambridge, England, UK), which was used for Aβ staining. Nuclei on the tissue were counterstained by Hoechst 33,342 (Invitrogen, Carlsbad, CA, USA). After the whole process, the specimen was covered by a cover glass with a mounting solution (HIGHDEF IHC mount, Enzo, Farmingdale, NY, USA). The sections were imaged using Axio Zoom microscopy (Carl Zeiss, Oberkochen, Baden-Württemberg, Germany). Quantification of hippocampus immunofluorescent images were conducted by ImageJ software [38]. The area percentage of fluorescence signal brighter than the local threshold was calculated compared to the whole tissue section area. More than 5 sections from each hemisphere sample with comparable morphology were selected for quantification. The antibodies are listed in Appendix A.

### 2.11. Statistical Analysis

Statistical analysis was performed with SigmaPlot software (Version 12.0). One-way ANOVA was used to determine the difference between experimental groups. *p* < 0.05 and *p* < 0.001 were considered significant.

## 3. Results

### 3.1. Mouse Induced Neural Stem Cells Expressing Spon1 Maintain Neural Stem Cell Characteristics

To investigate the ability of *Spon1* to reduce Aβ of autologous neural cells-mediated gene therapy, we generated induced neural stem cells (iNSCs) from autologous mouse fibroblasts by ectopic expression of *Oct4* through a direct conversion strategy as previously described [17] (Figure 1A). iNSCs were genetically engineered to express *Spon1* by transduction with a lentiviral vector encoding mouse *Spon1* coding sequence (iNSC-Spon1) (Figure 1B). We confirmed the integration of exogenous *Spon1* in iNSC-Spon1 and expression of exogenous *Spon1* in iNSC-Spon1 (Figure 1C,D). We also analyzed the endogenous *Spon1* expression. The expression level of endogenous *Spon1* in iNSC-Spon1 was similar to that of iNSCs (Figure 1E). The expression of exogenous Spon1 was also confirmed at the translational level by detecting a V5 tag linked to the 3′ terminal *Spon1* coding sequence (Figure 1F). To reveal whether the expression of exogenous *Spon1* did not change iNSC characteristics, the expression of NSC markers *Sox2, Pax6, Olig2*, and *Nestin* was confirmed by RT-PCR (Figure 1G). No significant difference in expression of these NSC markers was observed between iNSCs and iNSC-Spon1. We also confirmed that Nestin and Sox2 are expressed in iNSC-Spon1 through immunocytochemistry (Figure 1H). Thus, these results indicated that iNSCs can maintain their characteristics after engineering to express *Spon1*.

### 3.2. Mouse iNSCs Expressing Mouse Spon1 Reduce Amyloid Beta through the Bystander Effect

We confirmed the secretion of mouse SPON1 from iNSC-Spon1 since SPON1 is a secreted protein (Figure 2A). Mouse SPON1 (230 pg/mL) was detected after 24 h culture. To evaluate the effect of SPON1 on the production of Aβ40 and Aβ42 peptides, co-cultured media were collected after mouse Neuro 2a cells (N2a) were co-cultured with either iNSC-Spon1 or iNSCs for 48 h and quantified by ELISA. Both Aβ40 and Aβ42 concentrations decreased significantly when N2a was co-cultured with iNSC-Spon1 (49% and 37% compared to co-culture with iNSCs, respectively) (Figure 2B,C).

### 3.3. Human SPON1 Directly Reduces Amyloid Beta

Next, we evaluated the effect of *SPON1* by direct gene delivery. We engineered HEK 293T cells to express human *APP* with familial AD mutation by transducing a lentiviral vector encoding *APP* with Swedish (K670N/M671L), Florida (I716V), and London (V717I) mutations (293T-APP) (Figure 3A). 293T-APP highly expressed exogenous *APP* (Figure 3B) and secreted both Aβ40 (4053.9 pmol/L) and Aβ42 (794.6 pmol/L) (Figure 3C). We infected 293T-APP with a lentiviral vector encoding human *SPON1* (293T-APP-SPON1) (Figure 3D). We observed that exogenous V5-tagged SPON1 was expressed in 293T-APP-SPON1 and also confirmed the gene expression level of exogenous *SPON1* in 293T-APP-SPON1 by qRT-PCR (Figure 3E,F). There was no significant difference in the expression of genes (*APP*, *BACE1*, and *PSEN1*) related to amyloidogenesis in 293T-APP-SPON1 compared to 293T-APP (Figure 3F). We found that amount of Aβ40 (45%) and Aβ42 (24%) were lower in 293T-APP-SPON1 compared to 293T-APP (Figure 3H,I) thereby confirming that direct *SPON1* expression in 293T-APP affects the Aβ secretion. Thus, these results indicate that human *SPON1* is able to inhibit Aβ production.

### 3.4. Human SPON1 Induces Improvement of Learning in the Alzheimer’s Disease Model

To demonstrate that *SPON1* improves the cognition of AD model mice, we injected lentiviral human *SPON1* or *mKate2* (MOCK control) into both the hippocampus and entorhinal cortex (EC) of 5xFAD mice (three months old) at an early stage of AD pathological development (Figure 4A,B). Age-matched, non-transgenic wild-type mice underwent sham surgery as control.

Six months after *SPON1* delivery, mice (nine-month-old) were tested for learning and memory proficiency through the Morris water maze (MWM) assay. All mice were trained for five consecutive days (Figure 4C). The next day, probe trials were performed to confirm whether mice memorize the platform location from five days of training. (Figure 4D). Compared to mKate2-injected 5xFAD, *SPON1*-injected 5xFAD crossed the platform location often, spent more time in the target quadrant, and had a shorter mean distance to the target (Figure 4E–G).

To determine whether *SPON1* has an effect on the inhibition of Aβ production in vivo, we dissected the *SPON1*-injected 5xFAD brains and quantified Aβ by western blot. Soluble Aβ oligomers rather than plaques are known to be more cytotoxic and trigger synapse failure and memory impairment [39,40]. High molecular weight (~50 kDa) Aβ oligomers and 25 kDa Aβ oligomers were significantly reduced in the hippocampus region of *SPON1*-injected 5xFAD (Figure 4H–J). SPON1 was only detected in the *SPON1*-injected hippocampus. We could observe SPON1 expression only at the injection site in the hippocampus and EC of the brain section (Appendix A). We then conducted immunostaining of Aβ plaques to see further histological evidence of *SPON1* effect. The fluorescence signal of the *SPON1*-injected hippocampus and EC has less intensity and coverage of Aβ staining, compared to MOCK injected group (Figure 4K,L). The *SPON1*-injected hippocampus has significantly less Aβ plaques than the MOCK-injected hippocampus (Figure 4M). These data indicate that *SPON1* can reduce the Aβ and reverse cognitive impairment in the 5xFAD mouse model of AD.

## 4. Discussion

In this study, we demonstrated the effects of *SPON1* as a treatment option for AD. In particular, the direct injection of *SPON1* into the animal model of AD reduced Aβ levels and improved the memory function of the mice.

### 4.1. The Effect of SPON1 on Decreasing Aβ Secretion In Vitro

We observed that *SPON1* reduced Aβ levels both in the co-culture of mouse iNSC-Spon1 and mouse N2a cells and in human *SPON1* infection into human 293T-APP cells. Previous research from Ho et al. showed that SPON1 bound to the extracellular region of APP and inhibited BACE1 cleavage of APP [23]. In our study, Aβ40 and Aβ42 concentration were reduced in the co-culture of mouse iNSC-Spon1 and mouse N2a cells. This result indicates that the SPON1, which is secreted from iNSC-Spon1 cells, inhibits Aβ synthesis by binding to APP in the N2a cell membrane. We also confirmed that iNSC-Spon1 maintains NSC marker gene expression. These results suggest the feasibility of stem cell-mediated gene therapy of *SPON1* for AD patients. The direct conversion method allows to generate iNSCs from fibroblasts, which can be used for stem cell-mediated gene therapy. Moreover, we showed that Human *SPON1* also prevented the generation of Aβ40 and Aβ42 from human 293T-APP by viral gene transfer. These results indicate not only the therapeutic potency of human *SPON1*, but also the feasibility of human *SPON1* gene therapy in AD treatment.

### 4.2. Aβ Reduction and Reversal of Cognitive Impairment and Memory Dysfunction by Administration of SPON1 into 5xFAD Mice

We proved that *SPON1* injection ameliorated cognitive impairment and memory dysfunction in AD mice. Hafez et al. performed the behavior test after *Spon1* injection on AD mice but could not prove any improvement in memory and cognition behavior of AD mice [25]. The contradictory result from Hafez et al. might be the injection time of *Spon1*. They injected viral *Spon1* in 5.5-month-old AD mice even though Aβ began to deposit at three months of age. So far, one of the reasons for failure in AD therapies has been that the treatment begins after significant Aβ deposition has occurred. Recent studies have shown that treatment at an early stage of amyloid pathology was most effective [41,42]. Treatment with a γ-secretase inhibitor (GSI) for three months at four months of age resulted in a sustained reduction of amyloid plaque pathology in APP Tg2576 mice compared to treatment at seven or 12 months of age. Chronic treatments with a BACE1 inhibitor (GRL-8234) reduced cerebral Aβ42 levels and reversed memory impairments in four-month-old 5XFAD mice. In contrast, in 12-month-old 5XFAD mice, only marginal reductions of Aβ42 were observed after GRL-8234 treatment, and their memory function remained impaired. A previous study has shown that Aβ accumulation was dramatically increased between four and six months in 5xFAD [43]. In our experiments, virus-mediated *SPON1* was injected into young AD mice (three months old) to prevent Aβ plaque deposition from an early stage of amyloid pathology. We conducted the behavior test with nine-month-old mice, six months after *SPON1* injection, because previous studies established that nine-month-old 5xFAD mice showed a significant difference in learning and memory performance compared to their non-transgenic littermate [44,45].

In addition, Hafez et al. injected mouse *Spon1* into AD mice that have human APP transgene, but we have administered human *SPON1* to 5xFAD mice which express human *APP* transgenes. Since SPON1 binds to APP, matching the species would be necessary. Moreover, we decided to inject *SPON1* into both the hippocampus and entorhinal cortex (EC) to enhance hippocampal-dependent memory since the EC forms a neuronal network with the hippocampus and previous studies showed that brain-derived neurotrophic factor injection into the EC improved the hippocampal-dependent learning of the AD model [34,46].

Furthermore, we will also investigate the effect of *SPON1* on inflammation, neuronal death, and complement expression, which are phenotypes associated with aging in relatively old AD mice [47].

With the development of next-generation genome sequencing, PET imaging, and blood tests, AD patients can be diagnosed at an early stage of amyloid pathology [48,49,50]. *SPON1* gene therapy would be a promising option for AD treatment in early-stage amyloid pathology with the development of diagnostic technology.

## Figures and Tables

**Figure 1 cells-09-01275-f001:**
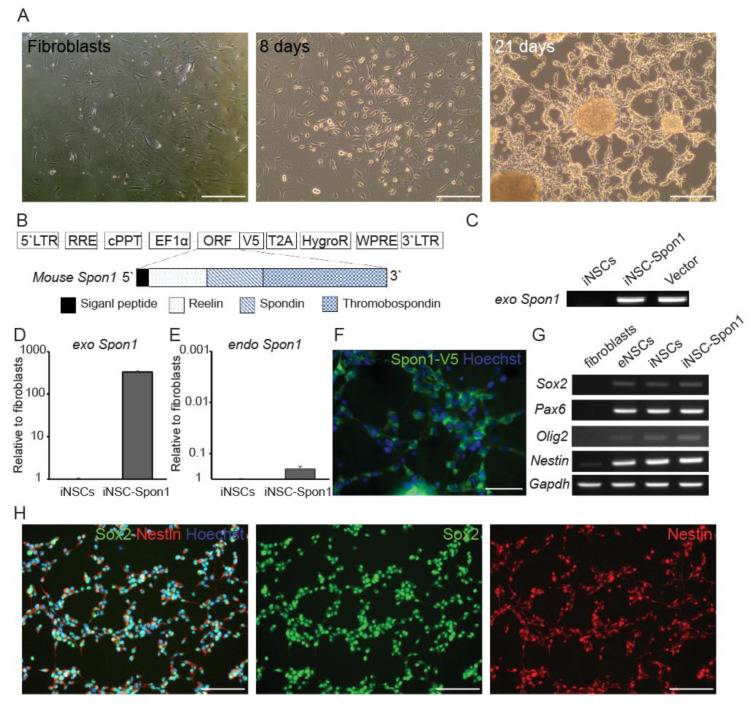
Generation of induced neural stem cells harboring *Spon1*. (**A**) Cell morphology changed after mouse *Octamer-binding transcription factor 4* (*Oct4*) infection. The left panel shows mouse fibroblasts used as parent cells before infection. The middle panel shows *Oct4*-infected cells 8 days after infection. The right panel shows NSC-like morphology 21 days after infection. Scale bar: 100 μm (**B**) Schematic drawing of lentiviral vector encoding full length of *Spon1*. A V5 tag was linked to the open reading frame (ORF). (**C**) Genotyping of integration of exogenous *Spon1* (exo *Spon1*) in iNSC-Spon1. (**D**,**E**) The expression of the mouse *Spon1* gene by qRT-PCR. Exogenous *Spon1* (exo *Spon1*) in iNSC-Spon1 (**D**). Endogenous *Spon1* (endo *Spon1*) gene expression was also quantified by qRT-PCR (**E**). Data are presented as mean ± SEM (*n* = 3). (**F**) Immunofluorescence images of ectopic expression of Spon1 in iNSC-Spon1. Anti-V5 antibody was used to detect V5-tagged Spon1. Scale bar: 50 μm (**G**) The expression of the NSC marker in engineered iNSCs using conventional PCR. Fibroblasts are parent cells of iNSCs and eNSCs are wild-type NSC. *Gapdh* is loading control. (**H**) Immunofluorescence images of NSC marker expression in iNSC-Spon1. Sox2 (Middle panel) and Nestin (Right panel) were co-stained and the nuclei were stained by Hoechst. The left panel is a merged image of Sox2, Nestin, and Hoechst. Scale bar: 100 μm.

**Figure 2 cells-09-01275-f002:**
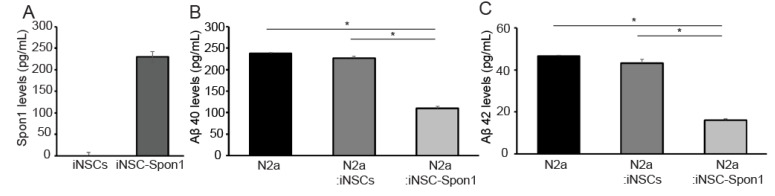
The reduction of Aβ by cell-mediated *Spon1* delivery. (**A**) The secretion of mouse SPON1 in iNSC-Spon1. The secretion amount from the cell culture medium was quantified by mouse SPON1 ELISA. Data are presented as mean ± SEM (*n* = 3). (**B**) The amount of secreted Aβ40 in the cultured medium was quantified by ELISA after co-culture of N2a with iNSC-Spon1. Data are presented as mean ± SEM (*n* = 3). * indicates *p* < 0.001 (One-way ANOVA). (**C**) The amount of secreted Aβ42 in the cultured medium was quantified by ELISA after co-culture N2a with engineered iNSCs. Data are presented as mean ± SEM (*n* = 3). * indicates *p* < 0.001 (One-way ANOVA).

**Figure 3 cells-09-01275-f003:**
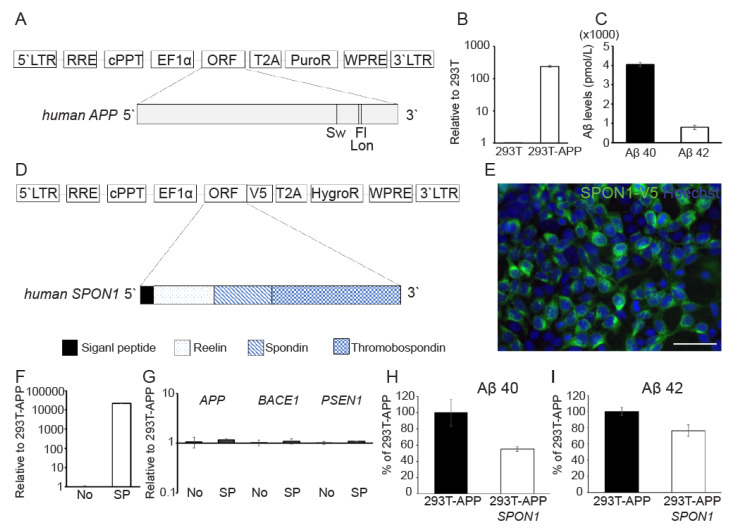
The reduction of Aβ by direct human *SPON1* gene delivery. (**A**) Schematic drawing of lentiviral vector encoding the human *APP* transgene with Swedish (K670N/M671L), Florida (I716V), and London (V717I) mutations. (**B**) Quantification of exogenous *APP* expression in 293T-APP using qRT-PCR. Data are presented as mean ± SEM (*n* = 3). (**C**) The quantification of secreted Aβ 40 and Aβ 42 in 293T-APP cultured medium by ELISA. Data are presented as mean ± SEM (*n* = 3). (**D**) Schematic drawing of lentiviral vector encoding full length of human *SPON1* coding sequences. V5 tag was linked to open reading frame (ORF). (**E**) Immunofluorescence image of ectopic expression of human SPON1 in 293T-APP-SPON1. Anti-V5 antibody was used to detect V5-tagged SPON1. Scale bar: 50 μm (**F**) Quantification of exogenous *SPON1* expression in 293T-APP (No) and 293T-APP-SPON1 (SP) using qRT-PCR. Data are presented as mean ± SEM (*n* = 3). (**G**) The expression of genes related to the amyloidogenic pathway in 293T-APP (No) and 293T-APP-SPON1 (SP) by qRT-PCR. Data are presented as mean ± SEM (*n* = 3). (**H**,**I**) The quantification of secreted Aβ 40 (**H**) and Aβ 42 (I) in 293T-APP and 293T-APP-SPON1 cultured medium by ELISA. The amount of secreted Aβ 40 or Aβ 42 was normalized to 100% in 293T-APP. Data are presented as mean ± SEM (*n* = 3).

**Figure 4 cells-09-01275-f004:**
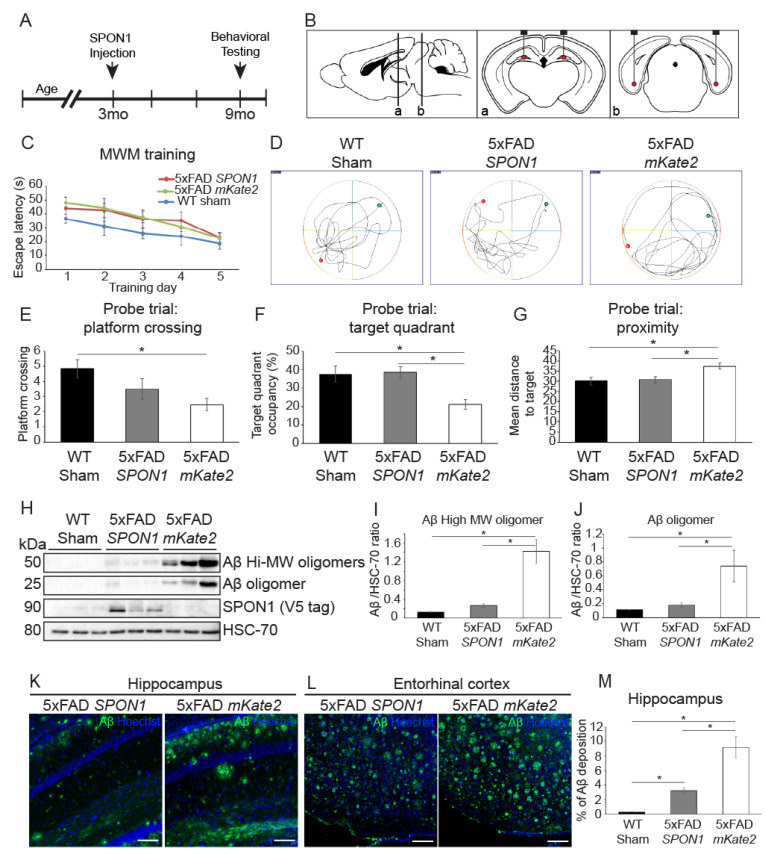
Alleviation of Spatial memory deficits in 5xFAD Alzheimer’s disease model mice by *SPON1*. (**A**) Schematic diagram of the experimental procedures. Intracranial injection of lentiviral vector encoding human *SPON1* coding sequences on 3-month-old mice. When the mice were nine months old, the behavioral (Morris water maze; MWM) test was performed. All mice were sacrificed after the behavioral test. (**B**) Schematic illustration of the four *SPON1* injection sites in the dentate gyrus of the hippocampus (a) and entorhinal cortex (b). (**C**) Escape latency in the MWM plotted against the training days. N number: WT sham 12, 5xFAD SPON1 14, 5xFAD mKate2 11. (**D**) Illustrative images of mouse trajectories during 1 min probe trial of the MWM test. The probe trial was performed without a platform 1 day after 5-day training. The platform was constantly located in the middle of the South/West during 5-day training. The green dot on the side of the North/East is the start site. The red dot is the last location of the mouse at the end of the 1-min probe trial. (**E**) Frequency of platform crossing during the 1-min probe trial of the MWM test. N number: WT sham 12, 5xFAD SPON1 14, 5xFAD mKate2 11. * indicates *p* < 0.05 (One-way ANOVA). (**F**) Percentage of time spent in the target quadrant during the 1-min probe trial of the MWM test. N number: WT sham 12, 5xFAD SPON1 14, 5xFAD mKate2 11. * indicates *p* < 0.05 (One-way ANOVA). (**G**) Mean distance to target during the 1-min probe trial of the MWM test. N number: WT sham 12, 5xFAD SPON1 14, 5xFAD mKate2 11. * indicates *p* < 0.05 (One-way ANOVA). (**H**–**J**) Aβ expression of 5xFAD by western blot. Western blot with the anti-Aβ (MOAB-2) antibody using the hippocampus from *SPON1* or mKate2-injected 5xFAD (5xFAD SPON1 or 5xFAD mKate2 respectively). Aβ high molecular weight (~50 kDa) oligomers and Aβ (~25 kDa) oligomer were detected. The density of bands was quantified by imageJ (**I**,**J**). V5-tagged SPON1 was detected by anti-V5 antibody. HSC-70 is shown as a loading control. * indicates *p* < 0.05 (One-way ANOVA). (**K**,**L**) Representative images of the hippocampus (**K**) and entorhinal cortex (**L**) showing Aβ plaques. Scale bar: 100 μm (**M**) Quantification of Aβ plaques in the hippocampus. Percentage of green fluorescence with respect to the area of the picture. * indicates *p* < 0.05 (One-way ANOVA).

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
