# Peer review of "SPON1 Can Reduce Amyloid Beta and Reverse Cognitive Impairment and Memory Dysfunction in Alzheimer’s Disease Mouse Model"

_cells, 2020, doi:10.3390/cells9051275_

Round 1
Reviewer 1 Report
//

Reviewer 2 Report
One possible therapeutic approach for AD is to target the enzymes that process amyloidogenesis. Recently, a direct lineage conversion was established to generate neural stem cells which proliferate well from fibroblasts by ectopic expression of the transcription factor Oct4 and then genetically engineered to express mouse Spon1 (iNSC-Spon1). When iNSC-Spon1 was co-cultured with Neuro 2a cells (N2a) which secretes Aβ, the amount of secreted Aβ was reduced. Therefore, the authors hypothesized that SPON1 may be used as a treatment for AD. To this end, human SPON1 virus was injected into hippocampus and entorhinal cortex of both hemispheres of 3 month-old 5xFAD female mice and Morris water maze (MWM) assay was performed to examine hippocampal-dependent learning and memory at 6 months after injection. The authors report that the direct injection of SPON1 into the animal model of AD improved the memory function of mice and reduced the Aβ levels. This is an interesting study. However, it is known that higher Aβ accumulation in the aged brain causes inflammation and neuronal death. Therefore the use of young mice to study an aging-associated disease might not be appropriate, especially in the inflammatory environment of the human aged brain (see, J Mol Med (Berl). 1995 Sep;73(9):465-71). This shall be acknowledged in the DISCUSSION.
Reviewer 3 Report
Article
SPON1 can Reduce Amyloid Beta and Reverse Cognitive Impairment and Memory Dysfunction in Alzheimer’s disease Mouse Model.
The authors showed the effect of SPON1 in Aβ reduction in vitro neural cells and in vivo AD mouse model. This study is interesting and new, the results are important, because SPON1 gene therapy would be a possible option for AD treatment.
However, this manuscript should be correct and improve before publishing:
General points:
Please add list of abbreviations to your manuscript.
Please improve the quality of the Figure 4 (K and L) and Supplementary Figure.
Special points:
Keywords: please also to keywords: amyloid beta; HEK 293T cells; Beta-secretase; induced neural stem cells.
Introduction
Lines 30-31: please add references at the end of each these three sentences.
Line 34: please add references at the end of this sentence.
Lines 35-36: please add references at the end of these each two sentences.
Line 37: please write out BACE1.
Line 40: please add references at the end of this sentence.
Line 41: please write out SPON1.
Line 47: please add references at the end of this sentence.
Line 49: please add references at the end of this sentence.
Line 50: please add references at the end of this sentence.
Line 53: please write out Oct4.
Line 57: please write out APP.
Line 58: please write out 5xFAD AD.
Materials and Methods
Line 63: please add more information about mouse primary dermal fibroblasts.
Line 75: please add more information about ATCC.
Line 106: please add which cells did you mean?
In vivo experiments
This section should be very significant improve.
Line 126: please add more information to Jackson Laboratory.
Surgical procedures:
Please add: Animals were housed in ? ; age of mice, weight of mice?
Please add: all conditions of the surgery: animals were deeply anesthetized via?
All surgeries were carried out?
Did you use a stereotactic frame? Which kind of frame?
The respective injection coordinates with reference to bregma were according to (please add references)?
Please add all very exactly about the injection: Human SPON1 virus (4 μl) was concentrated as previously described method [18] and stereotactically injected into hippocampus (AP: -2.02, ML: ± 1.8, DV: -1.9) and entorhinal cortex (AP: -4.7, ML: ± 2.9, DV: -3.1) of both side hemispheres of 3 month-old 5xFAD female mice (n=14).
Please add all very exactly about the Morris water mate test: Morris water maze (MWM) assay was performed to examine hippocampal-dependent learning and memory following standard protocol 6 months after injection [19,20]. The training for the hidden platform of the MWM assay consisted of four trials each day for 5 days. The probe trial was performed 24 hrs. after the last day of the training.
Line 134: mice were sacrificed.. How exactly? Please add to manuscript.
Line 149: please write out OCT.
Lines 150-151: The specimen was cryosectioned at 30 μm thickness. Using?
Lines 158-159: Quantification of hippocampus immunofluorescent images were conducted by ImageJ software. Please add more exacly information about ImageJ software
Discussion
Lines 298-308: please delete D.M.
Discussion should be intensive improve:
Please draw your Discussion section absolutely in the same way as your Results section using the same sections and then discuss all your results very exactly according all these sections.
Please describe very exactly the experiments and results from Hafez et al.. [24].
Please describe very exactly both references: [25,26].
Round 2
Reviewer 3 Report
Thank you, the manuscript was very intesive improved according to all my suggestions.